# Validity and reliability of the Swedish Functional Health Literacy scale and the Swedish Communicative and Critical Health Literacy scale in patients undergoing bariatric surgery in Sweden: a prospective psychometric evaluation study

Maria Jaensson ![ORCID],[1] Erik Stenberg ![ORCID],[2] Yuli Liang ![ORCID],[3] Ulrica Nilsson,[4] Karuna Dahlberg[1]

For numbered affiliations see end of article.

**Correspondence to**
Dr Maria Jaensson;
maria.jaensson@oru.se

## ABSTRACT

**Objectives** The aim was to psychometrically test and evaluate the Swedish functional health literacy scale and the Swedish communicative and critical health literacy scale in patients undergoing bariatric surgery.

**Design** A prospective cross-sectional psychometric study.

**Setting** Patients from three bariatric centres in Sweden were consecutively included in this study.

**Participants** A total of 704 patients undergoing bariatric surgery filled in the questionnaires preoperatively. Inclusion criteria were scheduled for primary bariatric surgery (Roux-en-Y gastric bypass or sleeve gastrectomy) and greater than 17 years, proficiency in Swedish.

**Primary and secondary measures** Psychometric outcomes of the Swedish Functional Health Literacy scale and the Swedish Communicative and Critical Health Literacy scale.

**Results** There was a higher proportion of females (74.4%, n=523) to males (25.6%, n=180). The mean age was 42 years (SD 11.5). Limited functional health literacy and limited communicative and critical health literacy (including both inadequate and problematic health literacy) was reported in 55% (n=390) and 40% (n=285), respectively. Cronbach alpha for the Swedish Functional Health Literacy scale was $\alpha$=0.86 and for the Swedish Communicative and Critical Health Literacy scale, $\alpha$=0.87. Construct validity showed weak to negative correlations between the Swedish Functional Health Literacy scale and income, education and SF-36/RAND36 summary scores. Confirmatory factor analysis showed a one-factor solution for the Swedish Functional Health Literacy scale and a two-factor solution for the Swedish Communicative and Critical Health Literacy scale.

**Conclusions** The Swedish Functional Health Literacy scale and the Swedish Communicative and Critical Health Literacy scale are valid and reliable to use for patients undergoing bariatric surgery in a Swedish context. Measuring dimensions of health literacy can be used as a guide for the development of health literacy friendly patient information in patients undergoing bariatric surgery.

## Strengths and limitations of this study

► This is the first psychometric evaluation in a Swedish sample undergoing bariatric surgery.
► This study contributes to the development of valid and reliable scales to be used in a clinical context.
► This study shows high prevalence of limited health literacy in patients undergoing bariatric surgery.
► The timing of preoperative data collection can be a limitation.
► There are few previous psychometric evaluations, making it difficult to compare results.

## INTRODUCTION

The concept of health literacy has been known for more than three decades.[1] Health literacy can be defined as 'a person's ability to assess, understand and use information to maintain or improve one's health'.[2] Health literacy can be divided into different skills or dimensions. Functional health literacy (FHL) covers basic skills in understanding and using health information. Communicative and critical health literacy (C & C HL) refers to more advanced social and cognitive skills in communication, applying and discriminating between different sources of information.[3 4] In recent years, research has investigated the consequences of limited health literacy for both the individual and the healthcare system. Factors such as lower educational level, lower household income, refugees, older age and sex are associated with limited health literacy.[5–10] Limited health literacy has been reported to be associated with poorer general health, poorer postoperative recovery and increased healthcare cost.[10–15] For the patient with limited health literacy, it can be

difficult to adhere to a medication regimen and to understand health information.[10] Of particular note is research that has shown that a patient with limited health literacy may ask fewer questions when meeting and talking to healthcare providers.[16] Knowledge of health literacy is therefore important for healthcare providers to provide equal care for all patients, especially when advocating a person-centred approach.[17]

Scales and instruments can measure health literacy, but there is no scale or instrument that is considered the gold standard. A recent systematic review found 18 different health literacy instruments used in surgical populations, for example, TOFHLA (test of FHL in adults), REALM (rapid estimate of adult literacy in medicine), NVS (newest vital sign), HALS (health and adult literacy survey), or BRIEF (brief health literacy screening tool)[18] This study uses two different scales measuring FHL, namely the Swedish FHL scale and the Swedish C & C HL scale. Both scales were originally developed and revised in Japan and have previously been translated to Swedish and are psychometrically tested and found to be valid and reliable in a Swedish context.[19–22] Exploratory factor analysis by Ishikawa *et al* supported a one-factor model for both scales.[19 23] The scales' psychometric properties have been evaluated in different populations, for instance, general population, patients with diabetes, refugees, patients undergoing day surgery and outpatient clinics attendees.[7 8 11 19 21 22]

The prevalence of limited health literacy in a variety of surgical patients has been shown in a meta-analysis to be greater than a third (34%, IQR 16%–50%).[18] Previous research, focusing on patients undergoing bariatric surgery, shows that 30%–50% of this population has limited health literacy.[24–26] While the link between socioeconomic factors and obesity remains complex, obesity appears to be more common for those living under less affluent circumstances in high-income countries, why it is likely that socioeconomic indifferences could contribute to the high rates of limited health literacy in this population[27] Bariatric surgery leads to a decrease in morbidity and mortality due to obesity and is the most effective method to reduce weight long term.[28 29] This surgery requires patients to adhere to several healthcare promotion guidelines to minimise surgical complications, lose excessive weight and learn healthier behaviour.[30] Therefore, patients are carefully prepared before surgery with education, among other things. However, analysis of patients' recall of information shows much information is forgotten or may not have been understood.[31] There is a risk, therefore, that patients with limited health literacy cannot fully adhere to desired healthcare promotion guidelines preoperatively and postoperatively with weight loss surgery.

When investigating a phenomenon such as health literacy, it is of utmost importance that the scale or instrument is psychometrically evaluated when used in different contexts, recognising that scales or instruments can be used to design patient information or education or to facilitate communication between healthcare provider and patient.[3 32] This study, therefore, aims to psychometrically test and evaluate the Swedish FHL scale, and the Swedish C & C HL scale in patients undergoing bariatric surgery.

## METHOD
### Study design
This psychometric evaluation was a cross-sectional survey conducted between October 2018 and November 2020 consisting of preoperative paper-based self-reported functional and communicative and critical health literacy in patients undergoing bariatric surgery. This study is part of a multicentre prospective, longitudinal mixed-methods study with an embedded design.[33]

### Sample and settings
Patients from three centres in Sweden were consecutively included in this study. The inclusion criteria were planned primary bariatric surgery (Roux-en-Y gastric bypass or sleeve gastrectomy) and greater than 17 years proficiency in Swedish.

All centres performing bariatric surgical procedures in Sweden report to the national quality and research register (Scandinavian Obesity Surgery Registry (SOReg)).[34] Data in the SOReg are collected at different timepoints: baseline (preoperatively and intraoperatively) and up to 10 years after surgery. For this psychometric study, cross-sectional data have been merged with data from SOReg collected at baseline before surgery.

Information about this study was given by a surgeon (in one centre) and a trained research nurse (in two centres) on a preoperative visit. If the patient agreed to participate, informed written consent was collected on the day of the surgery before filling in the study-specific form.

### Patient and public involvement
No patient involved.

### Data collection
To measure health literacy, the Swedish FHL scale and the Swedish C & C HL scale were used. Both scales consist of 5 items each. The Swedish FHL scale's first two items focus on visual ability related to the design of the text and its accessibility, the next two on the understanding of words and concepts, the fourth on perseverance in reading, and the last on the need for help in reading and understanding information. Responses are on a 5-point ordinal scale (1=never to 5=always) with lower scores indicating sufficient health literacy.[4]

The first three items in the Swedish C & C HL scale focus on the capacity for collecting, extracting, and understanding information related to health. The last two items focus on the abilities required to judge the reliability of the information and to apply health information to everyday life. Responses are on a 5-point ordinal scale

(1=strongly disagree to 5=strongly agree), with higher scores indicating a sufficient health literacy.[21 22]

Neither scale provides an initial summary score; instead, reported variables are re-categorised to three different levels (inadequate, problematic or sufficient health literacy) or to two categories (not inadequate/sufficient and inadequate) following guidelines instructions for the scales.[4 35]

To evaluate construct and discriminant validity, the following baseline variables were collected from SOReg: patient characteristics (age, gender, weight, body mass index (BMI) and type of surgery) and data from two additional scales, both of which have been found to be valid and reliable for Swedish contexts.[36–38] The first, Health-Related Quality of Life (HRQoL) 36-Item Short Form Health Survey, SF-36/RAND36 (hereafter referred to simply as SF-36), measures health-related quality of life and consists of 36 items grouped into eight scales (bodily pain, emotional role, general health, mental health, physical function, role function, social functioning and vitality). A higher score indicates better health status (ranging from 0 to 100). SF-36 provides two summary scores: the Physical Component Summary (PCS) and the Mental Component Summary (MCS) reflecting overall physical and mental health status.[36 37] The second, the obesity-related problem (OP) scale measures the impact of obesity on psychosocial functioning. It consists of eight questions on common OPs. A low score indicates better psychosocial functioning (ranging from 0 to 100).[38]

## Psychometric testing and statistical analysis

The psychometric evaluation was guided by the Consensus-Based Standards for the Selection of Health Measurement Instruments (COSMIN) guidelines.[39] There was no priori sample size calculation for this psychometric evaluation study.

Internal consistency measures the degree of interrelatedness and Cronbach alpha; coefficients $\geq 0.7$ and $\leq 0.95$ were regarded as acceptable.[40 41] Inter-item correlation was analysed to indicate if an item was part of the scale; appropriate values were 0.2–0.5.[42]

Floor and ceiling effects for each item were calculated as percentages of ratings at the lowest or highest level and was taken into account if more than 15% reported the lowest or the highest score.[41]

Construct validity was tested with predefined hypotheses drawn from previous research, namely that those participants with limited FHL have significantly lower educational level, lower income, and lower preoperative summary scores in PCS and MCS in SF-36.[9 10] When analysing correlations, the Swedish FHL scale and Swedish C & C HL scale were the dependent variables. To investigate differences between groups (health literacy and educational level and income), the $\chi^2$ test was used. Correlations were analysed using the Spearman r correlation test and were interpreted as follows: <0.1 was considered a negligible correlation, 0.1–0.39 weak correlations,

0.4–0.69 moderate correlations, 0.7–0.89 strong correlations and >0.89 very strong correlations.[43]

Structural validity was tested with confirmatory factor analysis (CFA).[39] We assumed a one-factor model with five items for the Swedish FHL scale and a two-factor model for the Swedish C & C HL scale.[44] Before running the CFA, the presence of outliers was investigated. Based on the squared Mahalanobis distance, there were 28 outliers and 29 outliers detected in the questionnaires of Swedish FHL and Swedish C & C HL, respectively. The CFA was conducted after removing the outlier observations. The following model fit criteria were used to evaluate the fit of the hypothetical model Comparative Fit Index (CFI) >0.9,[45] root mean square error of approximation (RMSEA) <0.08[46] and standardised root mean square residual (SRMR) <0.08.[47]

Discriminant validity was tested with the a priori hypotheses that the OP-scale has weak association (r≤0.39) with both the Swedish FHL and the Swedish C & C HL scale.[42]

Sociodemographic variables and health literacy were analysed descriptively and presented as number, percent, mean and SD.

FHL was categorised manually into sufficient, problematic and inadequate by recategorisation using the following transformations: 1 and 2 became 1; 3 became 100; and 4 and 5 became 1000. All values were summed and answers >1000 interpreted as inadequate, >100 and <1000 problematic, and <100 sufficient FHL.[4] C & C HL was categorised into sufficient, problematic and inadequate by recategorisation as follows; 1 and 2 became 1000; 3 became 100 and 4 and 5 became 1. All values were summed and totals ≥1000 interpreted as inadequate, >100 and <1000 as problematic and <100 as sufficient C & C HL.[35] Health literacy levels (both scales) were further dichotomised into limited (including inadequate/problematic) versus sufficient. Missing data were handled by listwise deletion.

The self-reported educational level was mapped to low and high educational level (compulsory school, upper secondary school, and higher vocational training vs first and second cycle education and third cycle study programmes).[48] Higher vocational training in Sweden lasts from 1 to 2 years.[49] The self-reported variable of income was mapped to low and high income (cut-off of 300 000 Swedish krona (SEK)/year, where €1 equals approximately SEK10.8 (June 2021)) and it was guided by the Statistics Sweden website which reports a median income of SEK337 000/year (2019) for men and women (20–64 years).[50]

A p<0.05 was considered significant. IBM SPSS Statistics, V.26, was used for the calculations. CFA was performed using the lavaan package, V.0.6–3.[51]

## RESULTS

A total of 704 patients gave their consent to participate in the study. Six forms were unit non-responses and three forms had partial responses in the Swedish FHL scale.

Seven forms were unit non-responses and 16 forms had partial responses in the Swedish C & C HL. Missing answers varied between 6 and 16 in each item in both scales. Most missing answers were in item four in Swedish C & C HL scale, 'Considering the credibility of the information.'.

Response rate for the different scales or instruments were 98.7% (n=695/704) for the Swedish FHL scale, 96.7% (n=681/704) for the Swedish C & C HL scale, and 72.3% (n=509/704) for the SF-36 and OP scales.

There was a higher proportion of females, 74.4% (n=523) to 25.6% (n=180) males. The mean age was 42 years (SD 11.5), and the mean preoperative weight and BMI were 120.2 kg/m$^2$ (SD 21.9) and 41.7 (SD 5.7), respectively. In this sample, 55% (n=390) reported limited FHL (including both inadequate and problematic FHL) and 40% (n=285) had limited C & C HL. Most patients reported upper secondary school as their highest level of education (52%, n=367/n=698). Almost 25% (24.6%) had undergone first, second or third cycle programmes at universities. The most frequent type of surgery was gastric bypass (56.8%, n=400) (table 1).

Item content and mean (SD) scores for the Swedish FHL scale and the Swedish C & C HL scale are seen in table 2.

### Floor ceiling effect/response distribution
There was a floor and ceiling for all items in both scales (table 2). The Swedish FHL scale had a floor effect with answers skewed to the left and between 24.6% and 61.4% of the answers to each item being 'never.' For the Swedish C & C HL scale answers were skewed to the right and questions were answered 'correct'. The ceiling effect varied between 27.6% and 66.9% for all items (table 2).

### Internal consistency
Cronbach alpha for the Swedish FHL scale was α=0.86 and for the Swedish C & C HL scale, α=0.87. Inter-item correlations varied between r=0.444 and r=0.702 (Swedish FHL scale) and r=0.434 and r=0.702 (Swedish C & C HL scale).

### Construct validity
Bivariate analyses showed significant associations with limited FHL for the independent variables of income (p=0.01) and educational level (p=0.02). Limited FHL was associated with lower income and lower educational level. There was negligible to weak negative correlations in all four a priori hypotheses (income, education, MCS and PCS) (table 3). There was no significant association between education and income with C & C HL. When analysing discriminant validity there was a negligible association between the OP scale and the Swedish FHL scale (r=0.054, p=0.22) and the C & C HL scale (r=0.046, p=0.31).

### Confirmatory factor analysis
The global fit can be assessed by the following criteria: CFI=0.996, RMSEA=0.043 and SRMR=0.013. The model shown in figure 1 had a very good model fit which

**Table 1** Sociodemographic variables, health literacy results and type of surgery

| Variables | |
|---|---|
| Sex female/male n (%) | 523 (74.4)/180 (25.6) |
| Age years M (SD) | 41.77 (11.4) |
| Weight (kg) M (SD)* | 120.27 (21.9) |
| BMI M (SD)* | 41.77 (5.7) |
| Level of educational n (%) | |
| Compulsory school | 67 (9.6) |
| Upper secondary school | 367 (52.6) |
| Other education (ie, higher vocational training) | 92 (13.2) |
| First and second cycle study programmes | 171 (24.5) |
| Third cycle study programmes | 1 (0.1) |
| Income (SEK) n (%) | |
| <SEK200 000 | 63 (9.1) |
| SEK200 000–SEK300 000 | 97 (13.9) |
| SEK300 000–SEK400 000 | 135 (19.4) |
| SEK400 000–SEK500 000 | 108 (15.3) |
| SEK500 000–SEK600 000 | 89 (12.8) |
| >SEK600 000 | 167 (24) |
| Unknown | 37 (5.3) |
| Swedish Functional Health Literacy n (%) | |
| Inadequate | 115(16) |
| Problematic | 275(39) |
| Sufficient | 305(43) |
| Swedish Communicative and Critical Health Literacy n (%) | |
| Inadequate | 45(6) |
| Problematic | 240(34) |
| Sufficient | 395(56) |
| Type of surgery n (%) | |
| Gastric bypass | 398 (56.5) |
| Sleeve gastrectomy | 294 (41.8) |
| Missing† | 10 |
| Other‡ | 2 (0.2) |

*n=698.
†Due to postponed or cancelled surgery.
‡Only diagnostic laparoscopy due to unexpected finding during surgery.
BMI, body mass index; SEK, Swedish kronor.

indicates that the one-factor solution is a nice fitting model of the Swedish FHL scale in patients undergoing bariatric surgery. Figure 1 shows the path diagram of the Swedish FHL as well as the factor loadings estimated by the maximum likelihood estimation method. The factor FHL explained almost half or more than half of the variances of items a, d, and e (0.46–0.63 with p<0.001) while

**Table 2** Item content, response distributions for each item, and mean (SD) and inter-item correlations for the Swedish FHL and the Swedish C & C HL scales

| Swedish FHL* | n | n (%) missing | Response category n (%) | | | | | Mean (SD) |
|---|---|---|---|---|---|---|---|---|
| | | | 1 | 2 | 3 | 4 | 5 | |
| 1.Do you think that it is difficult to read health information because the text is difficult to see (even if you have glasses or contact lenses)? | 697 | 7 (1) | 349 (49.6) | 187 (26.6) | 131 (18.6) | 25 (3.6) | 5 (0.7) | 1.78 (0.92) |
| 2.Do you think that it is difficult to understand word or numbers in health information? | 697 | 7 (1) | 269 (38.2) | 231 (32.8) | 164 (23.3) | 28 (4.0) | 5 (0.7) | 1.95 (0.91) |
| 3.Do you think that it is difficult to understand the message in health information? | 697 | 7 (1) | 263 (37.4) | 258 (36.6) | 153 (21.7) | 20 (2.8) | 3 (0.4) | 1.91 (0.86) |
| 4.Do you think that it takes a long time to read health information? | 698 | 6 (0.9) | 173 (24.6) | 216 (30.7) | 234 (33.2) | 57 (8.1) | 18 (2.6) | 2.33 (1.01) |
| 5.Do you ever ask someone else to read and explain health information? | 698 | 6 (0.9) | 432 (61.4) | 150 (21.3) | 76 (10.8) | 32 (4.5) | 8 (1.1) | 1.61 (0.93) |
| Swedish communicative and critical health literacy (C & C HL)† | n | n (%) missing | Response category n (%) | | | | | Mean (SD) |
| | | | 1 | 2 | 3 | 4 | 5 | |
| 1.Seeking information from various sources | 696 | 8 (1.1) | 2 (0.3) | 4 (0.6) | 36 (5.1) | 183 (26) | 471 (66.9) | 4.61 (0.63) |
| 2.Extracting relevant information | 696 | 8 (1.1) | 4 (0.6) | 17 (2.4) | 84 (11.9) | 259 (36.8) | 332 (47.2) | 4.29 (0.81) |
| 3.Understanding and communicating the information | 695 | 9 (1.3) | 3 (0.4) | 7 (1.0) | 112 (15.9) | 285 (40.5) | 288 (40.9) | 4.23 (0.76) |
| 4.Considering the credibility of the information | 688 | 16 (2.3) | 5 (0.7) | 19 (2.7) | 178 (25.3) | 292 (41.5) | 194 (27.6) | 3.95 (0.84) |
| 5.Making decisions based on the information | 693 | 11 (1.6) | 4 (0.6) | 6 (0.9) | 121 (17.2) | 320 (45.5) | 242 (34.4) | 4.14 (0.77) |

Swedish FHL scale: 1=never, 2=rarely, 3=sometimes, 4=often, 5=always. Swedish C & C HL scale: 1=not correct at all, 2=incorrect, 3=partly true, 4=correct, 5=exactly right.
*English translation of items by Mårtensson and Wångdahl.[4]
†English translation of items by Ishikawa et al.[19]
C & C HL, Communicative and Critical Health Literacy; FHL, Functional Health Literacy.

less than one-third of the variances of items b and c were explained by the factor FHL (0.27 and 0.33 with p<0.001). The factor loadings, which measured the correlations between a specific item and the FHL factor, ranged from 0.61 to 0.85 (p<0.001).

CFA showed a good fit for a two-factor model with correlation between two factors given in figure 2:

**Table 3** Spearman r correlations between the Swedish Functional Health Literacy scale and both the demographic characteristics and SF-36

| Variable | Value, r | P value |
|---|---|---|
| Education level | −0.122* | 0.003 |
| Income | −0.097† | 0.01 |
| SF-36 MCS | −0.251* | <0.001 |
| SF-36, PCS | −0.103 | 0.021 |

*Correlation is significant at the 0.001 level (two tailed).
†Correlation is significant at the 0.05 level (two tailed),.
HRQoL, Health-Related Quality of Life; MCS, Mental Component Summary; PCS, Physical Component Summary; SF-36, 36-Item Short Form Health Survey.

CFI=0.996, RMSEA=0.047 (90% CI 0.007–0.085) and SRMR=0.012. The value 0.88 on the curved double-headed arrows is the correlation between the two factors. Comparing this two-correlated-factor CFA model with the one-factor solution shows that the model in figure 2 has a better fit.

## DISCUSSION

This study shows that the Swedish FHL scale and the Swedish C & C HL scale are valid and reliable to use in patients undergoing bariatric surgery.

This study represents a sufficient sample size according to COSMIN guidelines[38] and also has high response rates with 695/704 (Swedish FHL scale) and 681/704 (Swedish C & C HL scale) responses.

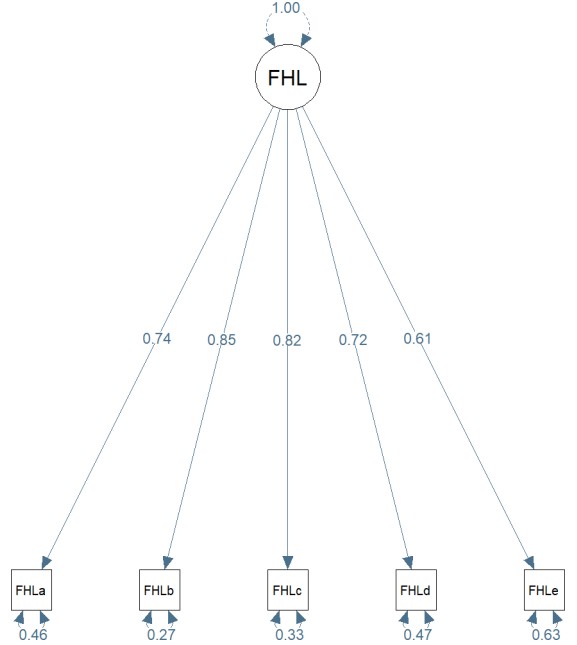

**Figure 1** Structural equation model for the Swedish FHL scale. Factor loadings were standardised. FHLa–e represent the questions in the Swedish Functional Health Literacy Scale. FHL, Functional Health Literacy.

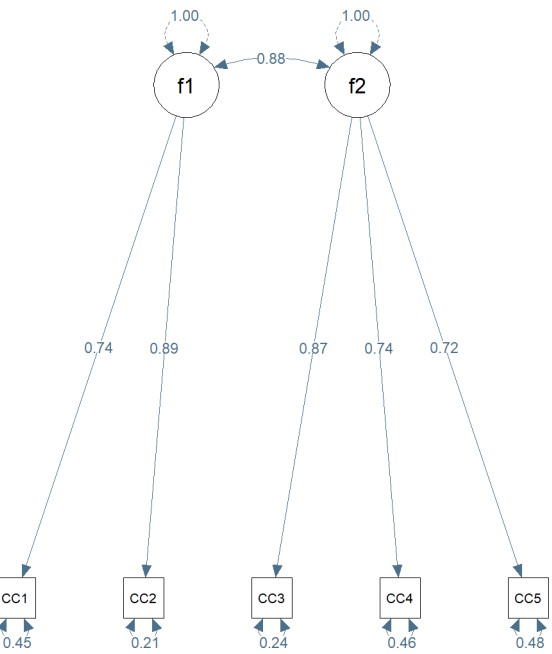

**Figure 2** Standardised parameter estimates for the two-factor model of Swedish C&C HL. F1 represents the communicative dimension and F2 represents the critical dimension. CC1-5 represent the questions in the Swedish C & C HL scale. C & C HL, Communicative and Critical Health Literacy.

When analysing the patients' responses, it was evident that both scales showed a skewed response distribution. This resulted from the large proportion of patients that reported answers on either the lower end (Swedish FHL scale) or the higher end (Swedish C & C HL scale) of the items. As far as we know, this response pattern has not been analysed in previous studies. The reason for this result in our study is not clear. It might be a characteristic of patients undergoing bariatric surgery. It might be a drawback of the construction of both scales.[52] However, the response scales in both the Swedish FHL scale and the Swedish C & C HL scale are categorical rating scales not different from other instruments measuring health literacy. For example, the HLS-EU-Q47 and the HL-EU Q16 have similar 4-point Likert scales with response options on a 4-point scale ranging from 'very difficult' to 'very easy.'[53]

One item stood out with the most missing answers (n=16), item 4 in the Swedish C & C HL scale, 'Considering the credibility of the information'. The reason for this is not clear though we did not perform a face or content validity assessment. One explanation might be that the patients found it difficult to value their own skill in knowing if information is credible or not. Responses to specific items are not to be found in previous psychometric evaluations,[20 22 23] suggesting that they should be studied further to see if it is an issue in other contexts as well.

Our analysis showed that both scales has good reliability with acceptable Cronbach alpha values of 0.86 (Swedish FHL scale) and 0.87 (Swedish C & C HL scale).[42] This

internal consistency is in line with recommended values.[40 41] Our result is also consistent with previous research on the FHL scale (Cronbach alpha of 0.684) and on the C & C HL (Cronbach alpha 0.86).[19 23] In addition, interitem correlations showed acceptable values, although one value of 0.702 indicates that the two items measure almost the same thing, with high interitem correlation values possibly showing that some items in the scales may be redundant.[42]

Construct validity was confirmed for the Swedish FHL scale by showing an association, although weak to negligible, with income, educational level and overall poorer health.[9 25 54] When interpreting these data, it is important to bear in mind that correlation coefficient association strengths are often interpreted literally, a misinterpretation that is problematic because associations do not show causal relationships.[43] Our results contribute to the growing body of evidence surrounding health literacy's impact on the individual as well as on the healthcare system.[3]

When applying CFA to investigate the factor structures for the scales, we obtained a one-factor model for FHL and a 2-factor model for C & C HL. One item (CC3) 'understanding and communicating the information' is now loaded on the critical dimension instead of the communicative dimension. This loading might be explained by the wording of this item, because 'understand' refers more strongly to critical aspects and people often share information with others after some critical analysis. Moreover, a better model fit was observed when CC3 was included in the critical dimension rather than the communicative dimension. Previous studies have also considered communicative and critical dimensions are two processing dimensions.[3 55]

The concept of health literacy has been known for decades and its importance to equality acknowledged.[56] Extensive research has shown challenges with limited health literacy not only for the individual but also for healthcare professionals. Despite extensive research on this topic, intervention to improve health literacy is lacking.[3] Some researchers state that when screening for health literacy there is a risk of stigmatisation, making the patient feel ashamed for not being able to understand and comprehend health information well enough.[57] Our result showed that 55% (n=390) reported limited FHL (including both inadequate and problematic FHL) and 40% (n=285) had limited C & C HL. Our results support the fact that there seems to be a higher prevalence of limited health literacy in patients with severe obesity undergoing bariatric surgery irrespective of country of origin.[24–26]

Despite the growing knowledge of the concept of health literacy there are very few interventional studies in clinical settings.[3] In the future a possible interventional study could use reliable and valid instruments to evaluate the effect of a potential intervention.

## Limitations

Despite the strengths of the multicentre design, with high inclusion rates and response rates, the study is not without limitations. The previous psychometric evaluations on these scales are quite sparse, making it difficult to compare results between studies. Therefore, psychometric evaluations need to be ongoing in different contexts to improve external validity on both scales. Both scales used in this study are shorter than other health literacy instruments; however, both scales are considered easy to use and patient friendly in a clinical context. The time required to fill out the forms before surgery might have had on impact on patients' answers. A better alternative might have been to collect data earlier in the preoperative period. According to the literature there is no consensus as to the optimal timing for the measurement of health literacy in a surgical population (preoperative or postoperative).[18]

Our data collection was performed in three centres that can be considered to be medium to high-volume centres operating mostly standard bariatric surgery as well as more advanced bariatric surgery. The samples represent the Swedish (and North European) average patient who undergo bariatric or metabolic surgery. Therefore, our results may not be generalised to other settings and countries.[58]

## CONCLUSION

The Swedish FHL scale and the Swedish C & C HL scale are valid and reliable instruments to use for patients undergoing bariatric surgery in a Swedish context. Measuring health literacy can be used as a guide for the development of health literacy friendly patient information in patients undergoing bariatric surgery.

**Author affiliations**
[1]Faculty of Medicine and Health, School of Health Sciences, Örebro University, Örebro, Sweden
[2]Department of Surgery, Faculty of Medicine and Health, Örebro university, Örebro, Sweden
[3]Department of Statistics, School of Business, Örebro University, Örebro, Sweden
[4]Department of Neurobiology, Care Sciences and Society, Karolinska Institutet and Perioperative Medicine, Karolinska University Hospital, Stockholm, Sweden

**Contributors** MJ: contributed to design, data collection, analysis and interpretation of data; wrote and critically revised the draft of manuscript; approved the final manuscript. ES: contributed to design, data collection, interpretation of data; critically revised manuscript and approved the final manuscript. YL: performed CFA and interpretation of data; wrote and critically revised the draft of the manuscript and approved the final manuscript. UN: contributed to design, interpretation of data; critically revised the manuscript and approved the final manuscript. KD: contributed to design, data collection, interpretation of data; critically revised the manuscript and approved the final manuscript. MJ: guarantor.

**Funding** This work was supported by Örebro University, grant number ORU 2018/00376 and ORU 2018/01219 and by ALF funding Region Örebro County (OLL-886141, OLL-935386 and OLL-939106) and Bengt Ihre Foundation (no grant number).

**Competing interests** None declared.

**Patient and public involvement** Patients and/or the public were not involved in the design, or conduct, or reporting, or dissemination plans of this research.

**Patient consent for publication** Not applicable.

**Ethics approval** The project was approved by the Regional Ethical Review Board in Uppsala, Sweden (ref: 2018/256) and followed good clinical practice and ethical principles as outlined by the 1964 Helsinki Declaration and its later amendments. All participants received written and oral information, including voluntary written consent and the option to withdraw at any time.

**Provenance and peer review** Not commissioned; externally peer reviewed.

**Data availability statement** No data are available. All data relevant to the study are included in the article or uploaded as online supplemental information.

**ORCID iDs**
Maria Jaensson http://orcid.org/0000-0001-7574-6745
Erik Stenberg http://orcid.org/0000-0001-9189-0093
Yuli Liang http://orcid.org/0000-0001-6581-7570

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
