## [Reviewer comments · BMJ Open]

ARTICLE DETAILS

TITLE (PROVISIONAL)	Validity and reliability of the Swedish Functional Health Literacy scale and the Swedish Communicative and Critical Health Literacy scale in patients undergoing bariatric surgery in Sweden: a prospective psychometric evaluation study
AUTHORS	Jaensson, Maria; Stenberg, Erik; Liang, Yuli; Nilsson, Ulrica; Dahlberg, Karuna

VERSION 1 – REVIEW

REVIEWER	Chun-Che Huang I-Shou University College of Medicine, Department of Healthcare Administration
REVIEW RETURNED	28-Sep-2021

GENERAL COMMENTS	This is an interesting and important topic, but there are several major and minor flaws in this study. Major: In the Methods section, 1) why the values of functional and communicative and critical health literacy were dichotomized into two categories? Why not to use three categories of health literacy levels? Please explain and provide supporting evidence. 2) the authors only reported the final sample and how many patients were excluded from study sample by these exclusion criteria? Please describe in detail. 3) the authors need to explain the values of Cronbach's alpha for the dimensions or each item. In the Result section, 1) I did not understand why the values of sample size was different from the description of the Abstract section. Please clarify this. 2) it is unclear why subjects with other type of surgery or missing were included according to the inclusion criteria? Please explain this discrepancy. 3) according to the results of Table 3, the weak negative correlations coefficient observed between the Swedish FHL scale and four variables (income, education, MCS and PCS). However, this study did not analyze the correlations between the Swedish C & C HL scale and four variables. Please add the analysis on this and explain. In the Discussion section, 1) data were only collected from the three centres which could limit the generalization of results to other medical settings or countries. 2) the inherent limitations of a cross-sectional design cannot be used to determine causal relationships. The authors need to add the limitations to the documentation. Minor: 1. Page 5, line 51: "Therefore patients are carefully prepared before surgery..." Please add the comma in this sentence "Therefore,
---

	patients are carefully prepared before surgery...". 2. Page 6, line 15: "This psychometric evaluation was a 'cross-sectional' survey..." Line 41: "For this psychometric study, 'cross sectional' data has been merged..." The authors should be amended consistently with the wording. 3. Page 7, lines 3-4: "Responses are on a 5-point ordinal scale (1=never to 5=always) with lower scores indicating..." Lines 11-12: "Responses are on a 5-point ordinal scale (1=strongly disagree to 5=strongly agree)", ' with higher scores indicating..." The authors should be amended consistently with the wording. 4. Page 7, line 29: "The second, the obesity-related problem (OP)scale measures..." Please add the space in this sentence "The second, the obesity-related problem (OP) scale measures..." 5. Page 7, line 31: "It consists of eight questions on common obesity-related problems." Please use "OP" abbreviations instead of obesity-related problems. 6. Page 7, lines 41-42: "...degree of interrelatedness and Cronbach alpha; coefficients ≥ 0.70 and ≤ 0.95 were regarded..." Please correct "0.70" to "0.7". In addition, lines 43: "...indicate if an item was part of the scale; appropriate values were 0.2 to 0.50.[41]" Please change "0.50" to "0.5". 7. Page 7, lines 57-59: "...were interpreted as follows: < 0.10 was considered a negligible correlation, 0.10-0.39 week correlations, 0.40-0.69 moderate correlations, 0.70-0.89 strong correlations,..." Please change "0.10", "0.40" and "0.70" to "0.1", "0.4" and "0.7". 8. Page 7, lines 58: "...0.10-0.39 week correlations,..." Please correct "week" to "weak". 9. Page 8, line 11: "...the fit of the hypothetical model comparative fit index (CFI) > 0.90 [44],..." Please change "0.90" to "0.9". 10. Page 8, line 15-16: "...the OP-scale has weak association ($r \leq 0.39$) with both the Swedish FHL..." Please correct "$r \leq 0.39$" to "$r \leq 0.39$".
--	--

REVIEWER	Daniel Chu The University of Alabama at Birmingham School of Medicine, Department of Surgery
REVIEW RETURNED	05-Oct-2021

GENERAL COMMENTS	Jaensson et al conducted a study assessing patient health literacy, communicative health literacy, and critical health literacy among Swedish patients undergoing bariatric surgery. They found that all three scales were valid and reliable to use for patients undergoing bariatric surgery. This study is well done and clearly validates the two instruments among the bariatric population. However, it is worth exploring the implications on patient care, comparison against other literacy instruments, and exploration of socioeconomic drivers. Altogether well done, but would benefit from additional discussion and description of the context and implications of the study.
--

	Major revisions  • The introduction mentions that there is no scale or instrument that is considered the gold standard, but does not mention TOFHLA (test of functional health literacy in adults), REALM (rapid estimate of adult literacy in medicine), NVS (newest vital sign), HALS (health and adult literacy survey), or BRIEF (brief health literacy screening tool). It would be topical to include these instruments. • What does it mean to ‘prepare patients with education’? • The study uses the FHL and C&C HI which have been psychometrically tested and found to be valid and reliable in a Swedish context. What concerns did the authors have that within a bariatric population specifically that they would no longer be valid and reliable? • The authors describe education level and income level in the results but not in the methods. • Do the patient demographics in this study reflect the general bariatric population in Sweden? Surprising to be 75% female. • How does this study compare to the validation of the instruments among Swedish populations as a whole? The reliability and Cronbach alpha values are similar, but are the skews observed similarly reflected in the original analysis? • Can the authors provide more detail about how this study contributes to the impact of health literacy on individual and healthcare systems? • The authors mention the stigma associated with identifying limited literacy, but don’t describe how to prevent the stigmatization of patients and how to use this identification strategy to improve and tailor their care. • The authors describe how the obesity is correlated with limited health literacy – is it socioeconomic in nature? Is obesity more so correlated with income, education level, or resource availability? Minor revisions  • Change loose excessive weight to lose excessive weight • Include IRB • Do the authors have access to data on how long it took to administer the instruments?
--	---

VERSION 1 – AUTHOR RESPONSE

Reviewer: 1

Dr. Chun-Che Huang, Kaohsiung Medical University Comments to the Author:

This is an interesting and important topic

Author comment: Thank you

*Why the values of functional and communicative and critical health literacy were dichotomized into two categories? Why not to use three categories of health literacy levels? Please explain and provide supporting evidence.

Author comment: Our analysis follows guidelines for use of the scales.

“The assessments can be divided into either three or two categories depending on which statistical calculations will be used. The numerical values are primarily meant for statistical calculations in the context of research.” (pp 5, <https://uploads.staticjw.com/ha/halsolitteracitet/guidelines-sfhl-swedish-version-3-1-january-2017.pdf>). Guidelines for S C& C HL is not translated to English but the instructions are that data can be divided to two or three categories.

This has been further clarified in data collection and it is supported by references (ref. 4 and 34)

*The authors only reported the final sample and how many patients were excluded from study sample by these exclusion criteria? Please describe in detail.

Author comments: We are not sure if we understand this comment correctly. The inclusion criteria’s were planned primary bariatric surgery (Roux-en-Y gastric bypass or sleeve gastrectomy) and greater

than 17 years and proficiency in Swedish. For this psychometric study, study forms that were not complete were excluded from the analysis. This is described in the beginning of the result.

“Six forms were unit nonresponses and 3 forms had partial responses in the Swedish FHL scale. Seven forms were unit nonresponses and 16 forms had partial responses in the Swedish C & C HL. Missing answers varied between 6 and 16 in each item in both scales.”

*The authors need to explain the values of Cronbach's alpha for the dimensions or each item.

Author comments: We interpret our Cronbach alpha values in the discussion. According to literature our Cronbach alpha coefficients are acceptable/good. We have not calculated Cronbach alpha for items or dimensions.

*I did not understand why the values of sample size was different from the description of the Abstract section. Please clarify this.

Author comment: A total of 704 patients were included in this study. One participant has missing on biological sex therefore there are 523 women and 180 men.

*It is unclear why subjects with other type of surgery or missing were included according to the inclusion criteria? Please explain this discrepancy.

Author comments: All patients included were considered for primary bariatric surgery. The inclusion was conducted before surgery. In 12 patients, surgery was postponed, cancelled or only a diagnostic laparoscopy was performed due to an unexpected finding during the operation. These patients cannot be evaluated for follow-up, but they still met the preoperative criteria for bariatric surgery and were therefore kept in the present study. This has been clarified in Table 1.

*According to the results of Table 3, the weak negative correlations coefficient observed between the Swedish FHL scale and four variables (income, education, MCS and PCS). However, this study did not analyze the correlations between the Swedish C & C HL scale and four variables. Please add the analysis on this and explain.

Author comments: We follow our published protocol for this project. Functional health literacy has previously been correlated with these four variables. We could not find any correlations between these variables and C & C HL in published literature. Therefore, we could not support any predefined hypotheses on previous literature. According to Cosmin guidelines it is of major importance that all hypotheses are defined in advanced when assessing construct validity. Therefore, we would prefer not to include the suggested analysis in the present study”

*Data were only collected from the three centres which could limit the generalization of results to other medical settings or countries.

Author comment:

All three centres can be considered to be medium to high-volume centres operating mostly standard bariatric surgery as well as more advanced bariatric surgery. The sample represents the Swedish (and North European) average patient who undergo bariatric or metabolic surgery. All three centres operate publicly funded operations, while one centre also includes a small proportion of privately funded operations. Thus, this also correlated well to the Swedish setting where the majority of operations are publicly funded. All patients were included consecutively and prospectively during the study period. We agree with the reviewer that the design was not that of an RCT, but in our opinion that would not be the optimal design given the research question. In response to the comment from the reviewer, we have expanded the discussion of generalizability.

*The inherent limitations of a cross-sectional design cannot be used to determine causal relationships. The authors need to add the limitations to the documentation.

Author comment: We do not determine causal relationships in this study. We merely evaluate these scales validity and reliability. We also discuss the limitations with interpretation on correlation in the discussion section “... a misinterpretation that is problematic because associations do not show causal relationships.[42]”.

*Page 5, line 51: “Therefore patients are carefully prepared before surgery...” Please add the comma in this sentence “Therefore, patients are carefully prepared before surgery...”.

Author comment: Amended

*Page 6, line 15: "This psychometric evaluation was a 'cross-sectional' survey..." Line 41: "For this psychometric study, 'cross sectional' data has been merged..." The authors should be amended consistently with the wording.

Author comment: Thank you for noticing this typo. It has been corrected.

*Page 7, lines 3-4: "Responses are on a 5-point ordinal scale (1=never to 5=always) with lower scores indicating..." Lines 11-12: "Responses are on a 5-point ordinal scale (1=strongly disagree to 5=strongly agree)", ' with higher scores indicating...' The authors should be amended consistently with the wording.

Author comments: These descriptions follow the guidelines for both scales. We are consistent describing scores indication sufficient functional or communicative and critical health literacy.

*Page 7, line 29: "The second, the obesity-related problem (OP)scale measures..." Please add the space in this sentence "The second, the obesity-related problem (OP) scale measures..."

Author comment: Thank you for noticing the missing space. It has been corrected

*Page 7, line 31: "It consists of eight questions on common obesity-related problems." Please use "OP" abbreviations instead of obesity-related problems.

Author comments: This has been corrected

*Page 7, lines 41-42: "...degree of interrelatedness and Cronbach alpha; coefficients ≥ 0.70 and ≤ 0.95 were regarded..." Please correct "0.70" to "0.7". In addition, lines 43: "...indicate if an item was part of the scale; appropriate values were 0.2 to 0.50.[41]" Please change "0.50" to "0.5".

Author comment: Thank you for noticing this typo. It has been corrected.

*Page 7, lines 57-59: "...were interpreted as follows: < 0.10 was considered a negligible correlation, 0.10-0.39 weak correlations, 0.40-0.69 moderate correlations, 0.70-0.89 strong correlations,..." Please change "0.10", "0.40" and "0.70" to "0.1", "0.4" and "0.7".

Author comment: Thank you for noticing this typo. It has been corrected

*Page 7, lines 58: "...0.10-0.39 weak correlations,..." Please correct "week" to "weak".

Author comment: Thank you for noticing this typo. It has been corrected

*Page 8, line 11: "...the fit of the hypothetical model comparative fit index (CFI) > 0.90 [44],..." Please change "0.90" to "0.9".

Author comment: Thank you for noticing this typo. It has been corrected

*Page 8, line 15-16: "...the OP-scale has weak association ($r \leq 0.39$) with both the Swedish FHL..." Please correct " $r \leq 0.39$ " to " $r \leq 0.39$ ".

Author comment: Thank you for noticing it has been corrected

Reviewer: 2

Dr. Daniel Chu, The University of Alabama at Birmingham School of Medicine Comments to the Author:

Jaensson et al conducted a study assessing patient health literacy, communicative health literacy, and critical health literacy among Swedish patients undergoing bariatric surgery. They found that all three scales were valid and reliable to use for patients undergoing bariatric surgery.

This study is well done and clearly validates the two instruments among the bariatric population. However, it is worth exploring the implications on patient care, comparison against other literacy instruments, and exploration of socioeconomic drivers. Altogether well done, but would benefit from additional discussion and description of the context and implications of the study.

Author comment: Thank you for your feedback

*The introduction mentions that there is no scale or instrument that is considered the gold standard, but does not mention TOFHLA (test of functional health literacy in adults), REALM (rapid estimate of adult literacy in medicine), NVS (newest vital sign), HALS (health and adult literacy survey), or BRIEF

(brief health literacy screening tool). It would be topical to include these instruments.

Author comment: Thank you for this suggestion. This has been added in the background section

*What does it mean to 'prepare patients with education'?

Author comment: Thank you for your question. This group of patients are going to group session with preoperative patient education about dietary instructions/restrictions, physiotherapy : exercise needed, information/education about the surgery, risk and benefit of the surgery. Also, one important key point is "this procedure is not a "magic pill" .

*The study uses the FHL and C&C HI which have been psychometrically tested and found to be valid and reliable in a Swedish context. What concerns did the authors have that within a bariatric population specifically that they would no longer be valid and reliable?

Author comments: Thank you for your question. To evaluate an instrument in different populations is considered a quality criterion in development and evaluation of an instrument/questionnaire/scale (Terwee et al, 2007). The scale used in this study were initially developed for a general population and further on patients with diabetes. By doing this psychometric evaluation we can therefore judge relevance and comprehensiveness in different items for these scales in a population of bariatric patients. We did find one item that seemed to be more difficult to respond to. Hopefully, more studies will expand our knowledge about the usefulness of these two scales in different populations.

*The authors describe education level and income level in the results but not in the methods.

Author comments: Educational level and income is described in the section Psychometric testing and statistical analysis ` (construct validity)

*Do the patient demographics in this study reflect the general bariatric population in Sweden? Surprising to be 75% female.

Author comment: The demographics reflects the general bariatric population in Sweden (Stenberg E, et al, Ann Surg 2014) as well as that seen in many parts of Europe. While obesity affects both men and women, globally most patients (74%) undergoing bariatric surgery are women (Welbourne R et al, Obes Surg 2019). In response to this question, and a question from reviewer #1 on generalizability, the discussion on this topic has been expanded in the discussion session"

*How does this study compare to the validation of the instruments among Swedish populations as a whole? The reliability and Cronbach alpha values are similar, but are the skews observed similarly reflected in the original analysis?

Author comments: This is an interesting question that we also are wondering about. Unfortunately, this has not been published in the original analysis, so we don't know the answer to this question.

*Can the authors provide more detail about how this study contributes to the impact of health literacy on individual and healthcare systems?

Author comment: This is going to be one of our focus in upcoming publications from this project, so therefore we do not want to expand on this in present manuscript. However, one aspect can be that patient education , as it is performed today, may be adapted and sustainable for all individuals regardless of health literacy level.

*The authors mention the stigma associated with identifying limited literacy, but don't describe how to prevent the stigmatization of patients and how to use this identification strategy to improve and tailor their care.

Author comment: This is a really interesting research question that we hope can be investigated further on.

*The authors describe how the obesity is correlated with limited health literacy – is it socioeconomic in nature? Is obesity more so correlated with income, education level, or resource availability?

Author comment: This is an interesting question addressing a complex issue. Although this was not the focus of the present study we agree with the reviewer that the correlation is of interest and should be the focus of upcoming studies. It is likely that socioeconomic differences contribute to the limited health literacy seen in this cohort. However, we find it likely that the correlation may be more complex

and multifactorial in nature. In response to this comment, the introduction has been revised to include a discussion on socioeconomic differences and obesity in high-income countries.

*Change loose excessive weight to lose excessive weight

Author comment: Thank you for noticing this typo. It has been corrected

*Include IRB

Author comment: The ethical application was reviewed by the Regional Ethics board in Uppsala which is our equivalence to the IRB. This was presented in the original manuscript (page 5, ethics paragraph)

*Do the authors have access to data on how long it took to administer the instruments?

Author comment: The clinical impression was that the instruments were easy and rather quick to complete. However, we did not record how long it took for each participant to fill out the instruments. Therefore, unfortunately, that information is not available.

VERSION 2 – REVIEW

REVIEWER	Chun-Che Huang I-Shou University College of Medicine, Department of Healthcare Administration
REVIEW RETURNED	05-Nov-2021
GENERAL COMMENTS	Thank you for the authors' responses and revision. I have no more questions about the revision.
REVIEWER	Daniel Chu The University of Alabama at Birmingham School of Medicine, Department of Surgery
REVIEW RETURNED	25-Oct-2021
GENERAL COMMENTS	Critiques addressed. Congrats to authors.